# Translation, cross-cultural adaptation, and psychometric evaluation of the Persian (Farsi) version of the QoLAF (quality of life in patients with anal fistula) questionnaire

Mohammad Reza Keramati[1,2]*, Seyed Mostafa Meshkati Yazd[1,2], Mostafa Omidi[1,2], Amir Keshvari[1,2], Sepehr Shahriarirad[3], Reza Shahriarirad[4,5], Seyed Mohsen Ahmadi-Tafti[1,2], Behnam Behboudi[1,2], Alireza Kazemeini[1,2], Leyla Sahebi[6], Mohammad Sadegh Fazeli[1,2]

**1** Division of Colorectal Surgery, Department of Surgery, Tehran University of Medical Sciences, Tehran, Iran, **2** Colorectal Research Center, Tehran University of Medical Sciences, Tehran, Iran, **3** Student Research Committee, Shiraz University of Medical Sciences, Shiraz, Iran, **4** Thoracic and Vascular Surgery Research Center, Shiraz University of Medical Sciences, Shiraz, Iran, **5** School of Medicine, Shiraz University of Medical Sciences, Shiraz, Iran, **6** Maternal, Fetal and Neonatal Research Center, Family Health Research Institute, Tehran University of Medical Sciences, Tehran, Iran

* mr-keramati@tums.ac.ir

**Data Availability Statement:** All relevant data are within the paper and its Supporting Information files.

## Abstract

### Introduction

The effective treatment of anal fistulas almost always requires surgical intervention, which could be accompanied by post-operative complications, and affect the quality of life of patients. This study aimed to cross-culturally adapt the Persian version of the Quality of Life in patients with Anal Fistula questionnaire and evaluate its validity and reliability.

### Materials and methods

Sixty patients with a mean age of 44 years ranging from 21 to 72 years entered the study. Forty-seven participants were men, and thirteen were women. After performing a scientific translation of the questionnaire based on Beaton's guidelines for cross-cultural adaptation and after extensive reviews by experts and specialists, the final version of the questionnaire was obtained. Then, 60 questionnaires (100%) were filled out by the participants (n = 60) and retrieved during a 7 to 21-day period. Data were collected and analyzed. Finally, according to the obtained data, the validity and reliability of the questionnaire were calculated.

### Results

Cross-cultural adaptation of the translated questionnaire was verified by the expert committee. The results showed perfect internal consistency (Cronbach alpha = 0.842), and external consistency (intraclass correlation coefficient = 0.800; P<0.001). Spearman correlation coefficient between test and retest was reported to be 0.980 (P-value <0.01), confirming the temporal stability of the translated questionnaire. The interrater reliability based on Cohen's

**Funding:** The authors received no specific funding for this work.

**Competing interests:** The authors have declared that no competing interests exist.

kappa coefficient also demonstrated a perfect degree of agreement between two peer variables (Kappa = 0.889; P<0.001).

## Conclusion

The Persian translation of the Quality of Life in patients with the Anal Fistula questionnaire was proven to be valid and reliable for the evaluation of the QoL of patients with anal fistula.

## 1 Introduction

Anal fistula (AF) is defined as an abnormal tract or cavity connecting the rectum or anal canal with the perianal skin [1]. Most AFs arise from cryptoglandular infections, and while this condition is generally considered to be an uncommon disease, it has an approximate prevalence of 17 cases per 100,000 patients [2–4]. Moreover, the average age onset of AF is between 20–50 years, and men are 2–3 times more likely to be affected [5]. The symptoms of this disease often include suppuration, anorectal pain, bleeding, and in some cases, difficulty controlling bowel movements [4, 6].

Considering the nature of AF and its adverse effect on a patient's quality of life (QoL), management of this disease has always required special attention. Treatment almost always requires surgical intervention with the goal being successful eradication of the fistula and preservation of anal continence [4, 7]. Despite most cases being easily treated via surgical operations, postoperative management, including improvement of healing time, fecal incontinence, and AF recurrence, the latest ranging from 10% to 57% [8], remains challenging [9]. Furthermore, a recent systematic review evaluated studies assessing treatment outcomes of patients with AF. While this study reported a substantial heterogeneity in outcomes and measurement instruments, the most reported outcomes included healing (77%), incontinence (63%), and recurrence (40%) [10].

In sequence, assessing patients' QoL, as an important indicator of treatment success, can provide valuable insight regarding surgical decisions and also the management of AF. Previously, certain general health-related QoL surveys, such as the Short Form 12 Health Survey (SF-12) and Short Form 36 Health Survey (SF-36) and also a variety of questionnaires related to patient incontinence, were used to provide a general evaluation of the QoL of patients with AF [11–16]. However, more recently, Ferrer-Márquez et al. developed a questionnaire that specifically measured the Quality of Life in patients with Anal Fistula (QoLAF) which was found to be valid and reliable(4). Afterwards, utilizing this newly developed questionnaire, Ferrer-Márquez et al. conducted another study evaluating the QoL of patients with AF where they found that AF exerts a moderate to high impact on the QoL of these patients [17].

To the best of our knowledge, since the development of the original QoLAF-Q, no other translation of the questionnaire has been made available. Therefore, this study aimed to cross-culturally [18] adapt a Persian (Farsi) translation of the Quality of Life in patients with Anal Fistula questionnaire (QoLAF-Q) and to evaluate the validity and reliability of this version of the QoLAF-Q.

## 2 Materials and methods

### 2.1 Study design

An observational cross-sectional study was conducted for the validation of the Persian (Farsi) QoLAF-Q. A Scientific translation was performed regarding the mentioned questionnaire, which was then approved by expert colorectal surgeons. 17 items were included in the

**Table 1. Item-total statistics of the Persian (Farsi) QoLAF-Q.**

|  | Scale mean if item deleted | Scale variance if item deleted | Corrected item-total correlation | Cronbach's Alpha if item deleted |
|---|---|---|---|---|
| Q1_test | 41.43 | 211.165 | .053 | .848 |
| Q2_test | 39.30 | 206.790 | .154 | .845 |
| Q3_test | 40.38 | 206.749 | .164 | .845 |
| Q4_test | 38.43 | 203.334 | .236 | .843 |
| Q5_test | 39.80 | 203.112 | .268 | .841 |
| Q6_test | 39.22 | 211.858 | .079 | .845 |
| Q7_test | 37.23 | 211.402 | .092 | .845 |
| Q8_test | 40.67 | 194.090 | .540 | .831 |
| Q9_test | 39.37 | 193.118 | .311 | .842 |
| Q10_test | 39.28 | 182.342 | .730 | .820 |
| Q11_test | 40.10 | 169.820 | .722 | .816 |
| Q12_test | 40.97 | 183.389 | .606 | .825 |
| Q13_test | 39.68 | 170.186 | .711 | .816 |
| Q14_test | 38.88 | 174.613 | .651 | .821 |
| Q15_test | 39.30 | 175.535 | .616 | .823 |
| Q16_test | 39.27 | 177.589 | .547 | .828 |
| Q17_test | 39.22 | 174.681 | .625 | .822 |

questionnaire with 11 items containing 6 domains (0 = very high, 1 = high, 2 = moderate, 3 = low, 4 = very low, 5 = none), 1 item containing 6 domains (0 = none, 1 = very low, 2 = low, 3 = moderate, 4 = high, 5 = very high), and 5 items containing 4 domains (0 = daily, 1 = weekly, 2 = monthly, 3 = never). Item-total statistics are provided in Table 1. The questionnaires were filled out by study participants, and the QoL of patients was divided into 3 groups based on the QoL scores: low QoL (0–25), intermediate QoL (26–50), and high (51–75).

## 2.2 Translation and cross-cultural adaptation

After acquiring permission from the owners of the original questionnaire [4] and based on the WHODAS 2.0 (WHO Disability Assessment Schedule) translation package [19], two Persian (Farsi) speaking researchers who were fluent in both Persian and English and well-aware of health and disability concepts and terms translated the questionnaire to Persian independently. The translators tested the translations for any discrepancies to reach a consensus and prepare the provisional Persian version. Then, the merged Persian version was translated back into English by another native English-speaking colleague who was also fluent in the Persian language and was blind to the original English terms and phrases of the questionnaire. The English back-translated version was then compared with the original version by the expert committee to make sure there were no conceptual or cross-cultural discrepancies [20]. Before constructing the final version of the questionnaire, preliminary testing was also performed with the help of the 10 patients with AF. The patients were asked to address any difficulties regarding the readability and understandability of the questionnaire. No concerns were reported during this phase [21].

## 2.3 Sample size calculation

The target sample size of 60 was calculated assuming 0.74 sensitivity, 0.94 specificity of the QoLAF questionnaire (in a pilot study), 0.3 prevalence of anal fistula surgery, 0.05 probability of type I error, and 0.2 precision [22].

## 2.4 Data gathering and participants

The study participants consisted of 60 adult patients who have recently been diagnosed with cryptoglandular AF at Tehran Imam Khomeini hospital in 2021. Demographic data, patient history, and specifications of the AF were noted. Further data was gathered using the finalized Persian version of the QoLAF-Q. The exclusion criteria for participants included IBD or any medical condition affecting patients' QoL, COPD, fibromyalgia, fistula secondary to carcinoma, radiotherapy, obstetric damage, or any sort of cognitive impairment affecting a patient's ability to properly fill out the questionnaire. Additionally, a certified psychologist expert in the field of psychosomatics evaluated and confirmed the patients to be mentally fit prior to their participation in the study. This study was approved by the Tehran University of Medical Sciences ethics committee (ethics code: IR.TUMS.IKHC.REC.1397.143). Additional ethical considerations regarding patients' anonymity were also appropriately met.

## 2.5 Statistical analysis

Gathered data were included in the IBM SPSS Statistics 26 edition software and were statistically analyzed. Mean ± SD was used for quantitative data, and qualitative data were displayed via frequency. A P-value of less than 0.05 was considered statistically significant. The post hoc power analysis was performed by G-power version 3.1.9.7 software, by bivariate normal model correlation from exact test family and 0.889 correlation between QoLAF and subjective self-reports of patients with anal fistula; the post hoc power was calculated as 0.88.

**2.5.1 Content validity and face validity.** Content validity is considered the degree to which an instrument has an appropriate sample size for a particular construct being measured [23, 24]. Face validity is a multidimensional construct useful in evaluating the accuracy and acceptance (or likability) of a certain test [25]. Evaluating these concepts, a panel of eight experts consisting of university faculty members reviewed the questionnaire to ensure equivalence in meaning during cross-cultural adaptation of the questionnaire. The panel shared their view on all items on the survey, and a 3-point Likert scale (poor, average, and good) was used for this purpose. Content validity of the Persian questionnaire was then confirmed and described as well based on the comments of the mentioned panel and previous experts. Regarding face validity, the patients were inquired to verify comprehension and readability of the questionnaire and agreed that the new questionnaire was understandable and easy to read for everyone.

**2.5.2 Convergent validity.** Convergent validity is considered as the substantial correlation between different instruments evaluating a common construct [26]. To assess convergent validity, we used Cohen's Kappa coefficient and Cross Tabulation to evaluate the level of concordance between QoLAF-Q scores and self-report results. For this purpose, the Persian translation of the question "Overall, how much has AF affected your quality of life? Mildly/Moderately/Severely" was previously added to the end of the questionnaire to provide a self-report of the overall effect of AF on the QoL of patients. This single QoL question has previously been used to for the development and validation of a questionnaire in a similar study [27, 28]. Based on the QoLAF-Q, three groups were defined: high QoL, intermediate QoL, and low QoL. A similar classification was also made for the subjective reports of the participants, including mild, moderate, and severe impactions. A possible correlation between these classifications was statistically analyzed, and based on the level of concordance, the convergent validity of the questionnaire was evaluated. If both the QoLAF-Q and self-report groups matched perfectly, a "Perfect fit" would be described. "Moderate fit" revealed a mismatch in one category, and "no fit" was used when the two groups were completely incompatible.

**2.5.3 Reliability.** To assess the reliability of the QoLAF questionnaire, four parameters were considered including internal consistency, external consistency, Spearman correlation

coefficient, and Cohen's kappa coefficient. A low value of Cronbach's alpha (<0.7) reveals a poor correlation between items and a high value is suggestive of redundancies. Thus, a value of alpha ranging from 0.7 to 0.9 would reveal ideal internal consistency [29], an intraclass correlation coefficient of at least 0.6 is considered to represent substantial reliability [30], and the value of Spearman's correlation coefficient ranges from -1 (perfect negative correlation) to +1 (perfect positive correlation) while a p-value of <0.05 represents a statistically significant association between analyzed variables [31], and for Cohen's Kappa coefficient $\kappa \leq 0.2$ is considered as slight, $0.2 < \kappa \leq 0.4$ as fair, $0.4 < \kappa \leq 0.6$ as moderate, $0.6 < \kappa \leq 0.8$ as substantial, and $0.8 < \kappa$ as almost perfect reliability, although a $\kappa$ of over 0.6 requires to be statistically significant in order to provide a definitive conclusion about the reliability of the measure [32]. A test-retest analysis was performed, and participants were contacted after 7 to 21 days to fill out the questionnaire again face-to-face after they had first completed it. The sample size did not suffer any changes during this phase. The two responses were then compared and gathered data were included in the software for statistical analysis.

## 3 Results

### 3.1 Participants and the QoL score

A total of 60 patients with a mean age of 44.1 (SD: 12.3, range: 21–72) years old joined the study. This population consisted of 43 (71.7%) men and 17 (28.3%) women. The average duration between the test and retest was 8.33 (SD: 2.79; range: 7–21) days. The mean QoLAF-Q score was 42 (SD: 14.6, range: 13–73). Based on QoLAF-Q scores, the QoL was determined to be low in 9 (15%), intermediate in 32 (53.3%), and high in 19 (31.7%) participants of the study (Table 2). On the other hand, 17 (28.3%) of the participants subjectively reported AFs to have little to no impact on their QoL, 30 (50.0%) reported moderate impaction, and 13 (21.7%) claimed that AFs had severely impacted their QoL (Table 2).

### 3.2 Consistency and test-retest reliability

All 60 participants of the study were inquired to fill out the QoLAF questionnaires a second time after 7 to 21 days. According to the second time's QoLAF questionnaire scores, 56 (93.3%) of the

**Table 2. Baseline features and the QoL reports among patients with anal fistula arriving at hospital during the year 2021.**

| Variables | | Value |
|---|---|---|
| Age (years); mean ± SD | | 44.1 ± 12.3 |
| Gender; n (%) | Male | 43 (71.7%) |
| | Female | 17 (28.3%) |
| Marital status; n (%) | Single | 7 (11.7) |
| | Married | 53 (88.3) |
| Duration between test and retest; mean ± SD | | 8.33 ± 2.79 |
| Average QoLAF-Q score; mean ± SD | | 42 ±14.6 |
| QoL score; n (%) | Low QoL | 9 (15%) |
| | Intermediate QoL | 32 (53.3%) |
| | High QoL | 19 (21.7%) |
| Subjective reports; n (%) | Mild impaction | 17 (28.3%) |
| | Moderate impaction | 30 (50.0%) |
| | Severe impaction | 13 (21.7%) |

SD: Standard deviation; QoL: Quality of life; QoLAF-Q: Quality of Life in patients with Anal Fistula questionnaire.

**Table 3. Cross tabulation between test and retest QoL scores in patients with anal fistula.**

| | | | Retest QoL score | | | |
|---|---|---|---|---|---|---|
| | | | Low QoL | Intermediate QoL | High QoL | Total |
| Test QoL score | Low QoL | Count | 8 | 1 | 0 | 9 |
| | | % within retest QoL score | 80.0% | 3.3% | 0.0% | 15.0% |
| | Intermediate QoL | Count | 2 | 29 | 1 | 32 |
| | | % within retest QoL score | 20.0% | 96.7% | 5.0% | 53.3% |
| | High QoL | Count | 0 | 0 | 19 | 19 |
| | | % within retest QoL score | 0.0% | 0.0% | 95.0% | 31.7% |
| Total | | Count | 10 | 30 | 20 | 60 |
| | | % within retest QoL score | 100.0% | 100.0% | 100.0% | 100.0% |

participants were placed in the same class as the first time, 4 (6.6%) changed by one category, and none (0.0%) were seen to change two categories. The results showed perfect internal consistency (Cronbach alpha = 0.842), and external consistency (intraclass correlation coefficient = 0.800; P<0.001). Also, Based on Spearman correlation coefficient analysis, there was a statistically significant difference between the first and second scores on the QoLAF questionnaire, indicating the reproducibility of the QoLAF-Q in different periods (Correlation coefficient: 0.980; P-value <0.001). The interrater reliability based on the Cohens kappa coefficient also demonstrated a perfect degree of agreement between two peer variables (Kappa = 0.889; P<0.001) (Table 3).

## 3.3 Convergent validity, sensitivity, and specificity

Convergent validity was evaluated by comparing QoLAF questionnaire scores (low, intermediate, high) and subjective self-reports of patients with AF (mild, moderate, severe) using Cohen's Kappa coefficient and Cross Tabulation, which showed a perfect agreement level (Kappa = 0.889; P-value< 0.001).

In 50 (83.3%) of the participants, a perfect fit was observed between QoL classifications based on questionnaire scores and self-reports. A moderate fit was observed in 10 (16.6%) of the participants, and none (0.0%) showed a significant difference between the two outcomes (Table 4). Therefore, the sensitivity and specificity of the QoLAF questionnaire were 61.5% and 94.1%, respectively.

## 3.4 Discriminative validity

According to these discriminative validity analyses, the QoLAF questionnaire was not able to differentiate between male and female participants (P-value = 0.978), nor could it classify participants based on their age (P-value = 0.287).

**Table 4. Cross tabulation between quality of life in patients with anal fistula questionnaire (QoLAF-Q) scores and quality of life self-reports of patients with anal fistula.**

| Quality of Life score | Total; N = 60 | Self-report impaction | | |
|---|---|---|---|---|
| | | Mild; n = 17 | Moderate; n = 30 | Severe; n = 13 |
| Low | 9 (15.0%) | 0 (0.0%) [c] | 1 (3.3%) [b] | 8 (61.5%) [a] |
| Intermediate | 32 (53.3%) | 1 (5.9%) [b] | 26 (86.7%) [a] | 5 (38.5%) [b] |
| High | 19 (31.7%) | 16 (94.1%) [a] | 3 (10.0%) [b] | 0 (0.0%) [c] |

a: Perfect fit

b: moderate fit

c: no fit

## 4 Discussion

The QoLAF-Q was originally designed in Spanish in 2017 with a group of 54 patients, which was also translated into English in the same study [4]. Since then, to the best of our knowledge, no translations of the questionnaire have been made available, which may partly be due to the small number of participants in the study developing the original questionnaire. Nevertheless, considering the benefits of having a more accurate tool for assessing the QoL of patients with AF, we transcribed a Persian version of the QoLAF-Q, which based on evaluations made in this study, the translation proved to contain internal and external consistency, good face and content validity, and its reliability was also confirmed.

Before the use of these surveys, reports had revealed a clear mismatch between what surgeons thought to be important QoL issues and how patients viewed the subject. An instance of this matter can be seen in a study where over 90% of surgeons designated continence as an important QoL factor, while only 25% of patients felt the same way [33]. With the development of various QoL surveys over the years, surgeons have been provided with decent tools to gain better insight into the QoL of patients with AF, affecting the process of decision-making as well as the management of post-operational events. Regarding these tools, Owen HA et al. conducted a study evaluating the QoL of patients with AF using St Mark's Incontinence Score which ranged from 0 (perfect continence) to 24 (totally incontinent) and Short Form 36 (SF-36) questionnaire, concluding that patients with AF suffer a reduced QoL, worsened in those with recurrent disease, secondary extension and urgency [11]. Also, in another study by Adamina M et al. the Fecal Incontinence Score Index and the Short Form-36 Health Survey,v2 (SF-36 v2) questionnaire were used to evaluate success, continence, and QoL in the treatment of complex fistulae [34]. While mentioned indices and questionnaires provide useful insight into the QoL of patients with AF, Ferrer-Márquezs et al. pointed out that these tools do not specifically evaluate the effect of AF on the QoL of these patients, thus developing a more specific, better-fitted questionnaire for this matter [4].

According to the QoLAF-Q scores in our study, approximately one-third of the participants (31.7%) were classified into the high QoL group, only 15% in the low QoL group, and about half of the participants (53.3%) made up the intermediate group. Also, when patients were asked to subjectively report the impact of the disease on their QoL, 28.3% reported mild impaction, 50.0% described moderate impaction, and 21.7% claimed they experienced a severe impaction on their QoL. On the other hand, quite similarly, in a study conducted by Ferrer-Márquezs et al in Spain, 35.1% of the participants were classified as high QoL, 36.3% as intermediate QoL, and 28.8% were classified as low QoL [17].

Assessing the reliability of the QoLAF-Q in our study, Cronbach's alpha was calculated to be 0.842, which validated the internal consistency of the questionnaire. Also, the Intra-class correlation coefficient was computed to be 0.80, confirming the questionnaire's external consistency as well. Next, temporal stability was assessed using Spearman's correlation coefficient between the test and retest phases of the questionnaire. The findings in this stage were statistically significant (P-value <0.01), rendering the Persian translation of the questionnaire reproducible and repeatable within different time frames. Finally, with test and retest, QoL scores as variables, Kappa's coefficient was 0.889, demonstrating a satisfactory level of concordance between the two tests.

In the reference study, a statistically significant correlation was detected between subjective reports of patients regarding the impact of AFs on their QoL and their classification based on questionnaire scores [4]. Similarly, in our study, after evaluating convergent validity using Cohen's Kappa coefficient and cross-tabulation between QoLAF-Q and subjective reports' scores, 83.3% of the participants showed a perfect fit between the two scores, 16,6%

demonstrated moderate fit, and none of the participants revealed any significant difference between the two tests. Comparing the reliability and validity scores of the Persian QoLAF-Q with the originally developed questionnaire [4], Cronbach's alpha coefficient was 0.842 in our study (vs. 0.908 in the reference study) which could be partially due to the increased length of our questionnaire (17-items vs. 14-items), increasing questionnaire reliability by shifting alpha to its ideal range and decreasing any redundancies [29]. Moreover, while Cohen's Kappa coefficient in our study was calculated to be 0.889 (p-value <0.001), Ferrer-Márquezs et al. also reported decent findings of quadratic κ = 0.82 (95% CI, 0.735–0.906) and linear κ = 0.72 (95% CI, 0.593–0.847). Spearman's correlation coefficient also showed near-perfect consistency in reproducibility of the questionnaires in both studies (0.980; p-value <0.001 vs. 0.861; p-value <0.001).

With the Persian version of the QoLAF-Q being statistically valid and reliable while also containing a satisfactory level of readability and understandability, this questionnaire can be considered for practical use to better evaluate the QoL of patients with AF while also providing better insight into the management of post-operational complications such as recurrence, incontinence, etc. Moreover, the use of a more specific tool for gauging the QoL of these patients can provide surgeons and researchers with valuable data, facilitating the evaluation of patient care while also making room for further comprehensive and in-depth research.

Some limitations need to be highlighted in our study. Considering the number of participants in our study (n = 60), a larger main sample would surely contribute to the production of more concrete results. The use of the single question "Overall, how much has AF affected your quality of life?" for the analysis of convergent validity is also another limitation of our study which was included to allow comparison with the reference study [4]. To the best of our knowledge, so far, no study has been conducted evaluating the effects of anal fistula on the QoL of the Iranian population. Also, according to a recent systematic review, all studies on the prevalence of anal fistula are related to Europe, and so far, no study has been conducted on other continents [35].

## 5 Conclusion

This study represents the Persian version of the QoLAF-Q has acceptable validity and perfect reliability in assessing the quality of life in Persian-speaking patients with anal fistula. In addition, this questionnaire is highly repeatable in different situations of time and place.

## Supporting information

**S1 File.**
(PDF)

**S2 File.**
(SAV)

## Author Contributions

**Conceptualization:** Mohammad Reza Keramati, Seyed Mostafa Meshkati Yazd, Mostafa Omidi, Amir Keshvari, Sepehr Shahriarirad, Reza Shahriarirad, Seyed Mohsen Ahmadi-Tafti, Behnam Behboudi, Alireza Kazemeini, Mohammad Sadegh Fazeli.

**Data curation:** Mohammad Reza Keramati, Seyed Mostafa Meshkati Yazd, Mostafa Omidi, Amir Keshvari, Sepehr Shahriarirad, Reza Shahriarirad, Seyed Mohsen Ahmadi-Tafti, Behnam Behboudi, Alireza Kazemeini, Mohammad Sadegh Fazeli.

**Formal analysis:** Mohammad Reza Keramati, Seyed Mostafa Meshkati Yazd, Mostafa Omidi, Amir Keshvari, Sepehr Shahriarirad, Reza Shahriarirad, Seyed Mohsen Ahmadi-Tafti, Behnam Behboudi, Alireza Kazemeini, Leyla Sahebi, Mohammad Sadegh Fazeli.

**Investigation:** Mohammad Reza Keramati, Seyed Mostafa Meshkati Yazd, Mostafa Omidi, Amir Keshvari, Sepehr Shahriarirad, Reza Shahriarirad, Seyed Mohsen Ahmadi-Tafti, Behnam Behboudi, Alireza Kazemeini, Mohammad Sadegh Fazeli.

**Methodology:** Mohammad Reza Keramati, Seyed Mostafa Meshkati Yazd, Mostafa Omidi, Amir Keshvari, Sepehr Shahriarirad, Reza Shahriarirad, Seyed Mohsen Ahmadi-Tafti, Behnam Behboudi, Alireza Kazemeini, Leyla Sahebi, Mohammad Sadegh Fazeli.

**Project administration:** Mohammad Reza Keramati, Seyed Mostafa Meshkati Yazd, Mostafa Omidi, Amir Keshvari, Sepehr Shahriarirad, Reza Shahriarirad, Seyed Mohsen Ahmadi-Tafti, Behnam Behboudi, Alireza Kazemeini, Mohammad Sadegh Fazeli.

**Resources:** Mohammad Reza Keramati, Seyed Mostafa Meshkati Yazd, Mostafa Omidi, Amir Keshvari, Sepehr Shahriarirad, Reza Shahriarirad, Seyed Mohsen Ahmadi-Tafti, Behnam Behboudi, Alireza Kazemeini, Mohammad Sadegh Fazeli.

**Supervision:** Mohammad Reza Keramati, Seyed Mostafa Meshkati Yazd, Mostafa Omidi, Amir Keshvari, Sepehr Shahriarirad, Reza Shahriarirad, Seyed Mohsen Ahmadi-Tafti, Behnam Behboudi, Alireza Kazemeini, Mohammad Sadegh Fazeli.

**Validation:** Mohammad Reza Keramati, Seyed Mostafa Meshkati Yazd, Mostafa Omidi, Amir Keshvari, Sepehr Shahriarirad, Reza Shahriarirad, Seyed Mohsen Ahmadi-Tafti, Behnam Behboudi, Alireza Kazemeini, Leyla Sahebi, Mohammad Sadegh Fazeli.

**Visualization:** Mohammad Reza Keramati, Seyed Mostafa Meshkati Yazd, Mostafa Omidi, Amir Keshvari, Sepehr Shahriarirad, Reza Shahriarirad, Seyed Mohsen Ahmadi-Tafti, Behnam Behboudi, Alireza Kazemeini, Mohammad Sadegh Fazeli.

**Writing – original draft:** Mohammad Reza Keramati, Seyed Mostafa Meshkati Yazd, Mostafa Omidi, Amir Keshvari, Sepehr Shahriarirad, Reza Shahriarirad, Seyed Mohsen Ahmadi-Tafti, Behnam Behboudi, Alireza Kazemeini, Mohammad Sadegh Fazeli.

**Writing – review & editing:** Mohammad Reza Keramati, Seyed Mostafa Meshkati Yazd, Mostafa Omidi, Amir Keshvari, Sepehr Shahriarirad, Reza Shahriarirad, Seyed Mohsen Ahmadi-Tafti, Behnam Behboudi, Alireza Kazemeini, Leyla Sahebi, Mohammad Sadegh Fazeli.

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
