## [Decision Letter · Decision Letter 0]

10 Aug 2022

PONE-D-22-15392Translation, Validation and Psychometric Evaluation of the Persian (Farsi) version of the QoLAF (Quality of Life in patients with Anal Fistula) questionnairePLOS ONE

Dear Dr. Keramati,

Thank you for submitting your manuscript to PLOS ONE. After careful consideration, we feel that it has merit but does not fully meet PLOS ONE’s publication criteria as it currently stands. Therefore, we invite you to submit a revised version of the manuscript that addresses the points raised during the review process.

 The external reviewer and I have now evaluated the manuscript. There are major points to be addressed before taking a positive decision on it. You could find the comments below. Please submit your revised manuscript by Sep 24 2022 11:59PM. If you will need more time than this to complete your revisions, please reply to this message or contact the journal office at plosone@plos.org. Please include the following items when submitting your revised manuscript:A rebuttal letter that responds to each point raised by the academic editor and reviewer(s). You should upload this letter as a separate file labeled 'Response to Reviewers'.A marked-up copy of your manuscript that highlights changes made to the original version. You should upload this as a separate file labeled 'Revised Manuscript with Track Changes'.An unmarked version of your revised paper without tracked changes. You should upload this as a separate file labeled 'Manuscript'.

We look forward to receiving your revised manuscript.

Kind regards,

Zubing Mei, MD,Ph.D

Academic Editor

PLOS ONE

Journal Requirements:

Reviewers' comments:

Reviewer's Responses to Questions

**Comments to the Author**

1. Is the manuscript technically sound, and do the data support the conclusions?

Reviewer #1: Partly

Reviewer #2: Yes

Reviewer #3: Yes

Reviewer #4: No

2. Has the statistical analysis been performed appropriately and rigorously? 

Reviewer #1: Yes

Reviewer #2: Yes

Reviewer #3: Yes

Reviewer #4: No

3. Have the authors made all data underlying the findings in their manuscript fully available?

Reviewer #1: Yes

Reviewer #2: Yes

Reviewer #3: Yes

Reviewer #4: Yes

4. Is the manuscript presented in an intelligible fashion and written in standard English?

Reviewer #1: Yes

Reviewer #2: Yes

Reviewer #3: Yes

Reviewer #4: Yes

5. Review Comments to the Author

Reviewer #1: PLOS ONE – PONE-D-22-00013

Decision: Major revision

Thank you for the opportunity to review this manuscript. In my opinion, the manuscript has some major corrections to be made and my comments are done section by section:

Abstract

# The introduction is too long and abbreviation should be avoided as much as possible

# Number of distributed and retrieved questionnaire was not mentioned.

# How was the translation scientifically done

Introduction

# there is inconsistency of references in the text

# there is a missing gap of how this disease affects the men and women in your locality

# how do they react when they are infected

Materials and Methods

# ‘the study participants consisted of 60 patients over the age 18 who have recently been diagnosed with crytoglandular AF…..’ the age is not consistent with what is stated in the abstract. Please reconcile

# in convergent validity, is there no reference(s) to support your assertion?

# ‘a test-retest analysis was performed and participants were contacted after 7 to 21 days to fill out the questionnaire again’. Were the participants certified fit psychologically before filling the questionnaire and how was the certification done?

Results

# remove grid lines in all the tables

# incomplete sentence in the first paragraph of convergent validity, sensitivity and specificity

References

@ no 8 reference has no page number. Kindly check

Reviewer #2: Dear authors,

I had the chance to read your article entitled "Translation, Validation and Psychometric Evaluation of the Persian (Farsi) version of

the QoLAF (Quality of Life in patients with Anal Fistula) questionnaire." You have evaluated the reliability and validity of the Persian version of this questionnaire. The article is well-constructed, and the English used in the article seems appropriate in most parts. However, I have some concerns, which are listed below.

1. Page 4, line 60, 'have been seem to be' seems crowded. Please change the sentence accordingly.

2. Did you perform preliminary testing before constructing the final version of the questionnaire?

3. Please briefly clarify how the readers should interpret the values for Cronbach alpha, ICC, spearman's rho, etc.

4. How did you contact the patients after 7-21 days for a retest? By telephone, e-mail, or face-to-face?

5. I prefer seeing the means of retest scores in a separate table with the statistical results. I recommend you construct another table for this purpose.

6. In discussion, a paragraph containing the head-to-head comparison of reliability and validity scores of statistical analysis with the original questionnaire (e.g., Cronbach alpha scores, ICC scores) is necessary.

7. Doesn't your study have any limitations? Please add a limitations paragraph.

Reviewer #3: Dear Authors,

I want to appreciate you for drafting this manuscript. It was well written and needed for the population.

However, there is a need for adjustments to the manuscript's structure. Firstly, I will suggest the titled be changed to “Translation, Cross-cultural Adaptation, and Psychometric evaluation of the Persian (Farsi) version of the QoLAF (Quality of Life in patients with Anal Fistula) questionnaire. Secondly, the following should be noted in the abstract:

1. The study did not capture which specific translation process used. It should be mentioned if it was based on the WHO translation process or any other that was specifically followed. The assessment of the manuscript would be based on which specific guideline that was used.

2. The results section should briefly capture the translation, what was found during the cross-cultural adaptation (CCA) and major psychometrics like validity and reliability.

3. There are too many unnecessary results in the abstract.

Finally, the authors should be able to distinguish between the three parameters of cross-cultural adaptation of a tool and present their results separately; 1. Translation 2. Adaptation 3. Psychometric property testing.

A clear distinction should be made between translation, adaptation and cross-cultural

validation. The translation is the single process of producing a document from a source version

in the target language. Adaptation refers to the process of considering any differences

between the source and the target culture to maintain equivalence in meaning. This

adaptation is referred to CCA. The cross-cultural validation of a questionnaire is a different process from the CCA. Cross-cultural validation aims to ensure that the new questionnaire functions as intended, has the same properties as the original, and functions similarly.

Kindly note that no results of the adaptation were presented in the manuscript.

A clear distinction should be made between translation, adaptation and cross-cultural

validation. The translation is the single process of producing a document from a source version

in the target language. Adaptation refers to the process of considering any differences

between the source and the target culture to maintain equivalence in meaning. This

adaptation is referred to CCA. The cross-cultural validation of a questionnaire is different

process from the CCA. Cross-cultural validation aims to ensure that the new questionnaire

functions as intended and has the same properties as the original and functions in the same

way.

Reviewer #4: The authors conducted a cross-sectional study on translation, validation and psychometric evaluation of the Persian (Farsi) version of the QoLAF (Quality of Life in patients with Anal Fistula) questionnaire in a sample of patients with AF. The results showed that the Persian version of QoLAF is valid and reliable. However, there are some issues that the authors should be addressed before publication.

- Introduction:

o It would be nice if the authors more deeply refer to previous studies on outcomes of patients with AF.

o Are there available the other translated versions of the QoLAF?

- Methods

o The method section should be written in more details.

o Line 84: this is a validation study not the developmental one.

o Please describe that which international method was used for translation and validation process.

o In the method section, there is no information regarding the QoLAF. The authors should provide full description about the questionnaire including the number of items, number of domains (if any), and the scoring system.

o Please clarify that who the authors reached a sample size of 60 cases.

o The statistical analysis section should have sufficient information for reviewers and readers to be able to determine that the methodology is sound and valid for the planned analyses. This section also needs to provide information, such as a power analysis to support the accrual number request, which level of reliability and validity are clinically acceptable. The number of planned subjects to enroll should be adequate to provide sufficient data for validity and reliability results.

o Please provide the definition of content and convergent validities with supporting references. I strongly suggest the authors to read the following articles before revising the method section of this manuscript.

“Assessment of reliability and validity of the adapted Persian version

of the Spinal Appearance Questionnaire in adolescents with idiopathic

scoliosis. Spine Deform. 2022;10(2):317-26.”

“Guidelines for the process of cross-cultural adaptation of self-report measures.

Spine (Phila Pa 1976) 25(24):3186–3191”

“Quality criteria were proposed for measurement properties of health status questionnaires. J Clin Epidemiol 60(1):34–42”

o Line 125: what are the self-report scores?

o What was the sample size for test-retest reliability?

Results:

o Table 1: This table is not provided in a standard form.

o Line 155: Heading of “Consistency and reliability” should be written as “consistency and test-retest reliability”

- Discussion:

o In the last paragraph of the discussion section, please provide the study limitations.

6. PLOS authors have the option to publish the peer review history of their article (what does this mean?). If published, this will include your full peer review and any attached files.

Reviewer #1: **Yes: **CHINEDU GODWIN UZOMBA

Reviewer #2: No

Reviewer #3: **Yes: **Prof Bashir Bello

Reviewer #4: No

---

## [Author Response · Author response to Decision Letter 0]

21 Sep 2022

Reviewer #1

Comment 1: Abstract: The introduction is too long and abbreviation should be avoided as much as possible.

Authors’ response: Thank you very much for your valuable comments. Abstract has been summarized and revised accordingly.

Comment 2: Abstract: Number of distributed and retrieved questionnaire was not mentioned.

Authors’ response: Thanks. Requested data has been added to the abstract.

Comment 3: Abstract: How was the translation scientifically done

Authors’ response: Thanks. The translation was performed based on the WHODAS 2.0 translation package. Manuscript (the Translation part of the Methods section of the manuscript) has been updated accordingly.

Comment 4: Introduction: There is inconsistency of references in the text

Authors’ response: Thanks. The references have been checked to comply with the journal’s guidelines (Vancouver style) and in numerical order. If the honorable reviewer has a specific reference in mind, we would gladly amend based on their recommendation.

Comment 5: Introduction: There is a missing gap of how this disease affects the men and women in your locality. How do they react when they are infected.

Authors’ response: Thanks. To the best of our knowledge, so far no study has been conducted evaluating the effects of anal fistula on the QoL of the Iranian population. Also, according to a recent systematic review, all studies on the prevalence of anal fistula are related to Europe, and so far, no study has been conducted on other continents. Thus, to portray the overall effect of anal fistula on patients, the results of a recent systematic review has been added to the manuscript. 

Comment 6: Materials and Methods: ‘the study participants consisted of 60 patients over the age 18 who have recently been diagnosed with crytoglandular AF…..’ the age is not consistent with what is stated in the abstract. Please reconcile

Authors’ response: Thanks. Revision has been made accordingly to better clarify the included participants. 

Comment 7: Materials and Methods: in convergent validity, is there no reference(s) to support your assertion?

Authors’ response: Thanks. Updates to the manuscript have been made and relevant references were added accordingly.

Comment 8: Materials and Methods: ‘a test-retest analysis was performed and participants were contacted after 7 to 21 days to fill out the questionnaire again’. Were the participants certified fit psychologically before filling the questionnaire and how was the certification done?

Authors’ response: Thanks. A certified psychologist and expert in the field of psychosomatics evaluated and confirmed the patients to be mentally fit prior to their participation in the study. Manuscript has been updated accordingly.

Comment 9: Results: Remove grid lines in all the tables

Authors’ response: Thanks. The grid lines have been removed as requested.

Comment 10: Results: incomplete sentence in the first paragraph of convergent validity, sensitivity and specificity

Authors’ response: Thanks. The mentioned sentence has been completed.

Comment 11: References: no 8 reference has no page number. Kindly check

Authors’ response: Thanks. The mentioned reference and also all other references have been checked and updated.

Reviewer #2 

Comment 1: Page 4, line 60, 'have been seem to be' seems crowded. Please change the sentence accordingly.

Authors’ response: Thank you very much for your valuable comments. Sentence has been revised accordingly.

Comment 2: Did you perform preliminary testing before constructing the final version of the questionnaire?

Authors’ response: Thanks. Preliminary testing was performed with the help of 10 patients with AF who were asked to inform us of any difficulties in readability or understandability of the questionnaire. No concerns were reported during this phase. The manuscript has been updated accordingly in the Materials and Methods section.

Comment 3: Please briefly clarify how the readers should interpret the values for Cronbach alpha, ICC, spearman's rho, etc.

Authors’ response: Thanks. In the Materials and Methods section a brief description addressing requested subject has been added to the manuscript. 

Comment 4: How did you contact the patients after 7-21 days for a retest? By telephone, e-mail, or face-to-face?

Authors’ response: Thanks. The retest was performed face-to-face. Addition to the manuscript (Reliability part of the Materials section) has been made accordingly.

Comment 5: I prefer seeing the means of retest scores in a separate table with the statistical results. I recommend you construct another table for this purpose.

Authors’ response: Thanks. New table consisting of retest scores with statistical results has been added to the Materials section as requested.

Comment 6: In discussion, a paragraph containing the head-to-head comparison of reliability and validity scores of statistical analysis with the original questionnaire (e.g., Cronbach alpha scores, ICC scores) is necessary.

Authors’ response: Thanks. Reliability and validity scores have been compared in a newly added paragraph as encouraged. 

Comment 7: Doesn't your study have any limitations? Please add a limitations paragraph.

Authors’ response: Thanks. Limitations paragraph added at the end of the Discussion section as requested.

Reviewer #3

Comment 1: Firstly, I will suggest the titled be changed to “Translation, Cross-cultural Adaptation, and Psychometric evaluation of the Persian (Farsi) version of the QoLAF (Quality of Life in patients with Anal Fistula) questionnaire. 

Authors’ response: Thank you very much for your valuable comments. The title has been changed accordingly.

Comment 2: Abstract: The study did not capture which specific translation process used. It should be mentioned if it was based on the WHO translation process or any other that was specifically followed. The assessment of the manuscript would be based on which specific guideline that was used.

Authors’ response: Thanks. The translation was performed based on the WHODAS 2.0 translation package. Manuscript (Materials and Methods section) has been revised accordingly.

Comment 3: Abstract: The results section should briefly capture the translation, what was found during the cross-cultural adaptation (CCA) and major psychometrics like validity and reliability.

Authors’ response: Thanks. Abstract has been revised accordingly.

Comment 4: There are too many unnecessary results in the abstract.

Authors’ response: Thanks. Abstract has been summarized and revised accordingly.

Comment 5: Finally, the authors should be able to distinguish between the three parameters of cross-cultural adaptation of a tool and present their results separately; 1. Translation 2. Adaptation 3. Psychometric property testing. A clear distinction should be made between translation, adaptation and cross-cultural validation. The translation is the single process of producing a document from a source version in the target language. Adaptation refers to the process of considering any differences between the source and the target culture to maintain equivalence in meaning. This adaptation is referred to CCA. The cross-cultural validation of a questionnaire is a different process from the CCA. Cross-cultural validation aims to ensure that the new questionnaire functions as intended, has the same properties as the original, and functions similarly. Kindly note that no results of the adaptation were presented in the manuscript.

Authors’ response: Thanks. Manuscript has been revised accordingly to better capture cross-cultural adaptation and validity of the translated questionnaire.

Reviewer #4

Comment 1: Introduction: It would be nice if the authors more deeply refer to previous studies on outcomes of patients with AF.

Authors’ response: Thank you very much for your valuable comments. Findings of a number of additional studies have been added to the manuscript as suggested.

Comment 2: Introduction: Are there available the other translated versions of the QoLAF?

Authors’ response: Thanks. To the best of our knowledge, no other translations of the QoLAF-Q have been made available since the development of the original questionnaire. A sentence conveying this message has been added to the introduction section.

Comment 3: Methods: The method section should be written in more details. Line 84: this is a validation study not the developmental one.

Authors’ response: Thanks. Further revision of the methods section has been conducted accordingly.

Comment 4: Methods: Please describe that which international method was used for translation and validation process.

Authors’ response: Thanks. The translation was performed based on the WHODAS 2.0 translation package.

Comment 5: Methods: In the method section, there is no information regarding the QoLAF. The authors should provide full description about the questionnaire including the number of items, number of domains (if any), and the scoring system.

Authors’ response: Thanks. Manuscript has been updated accordingly.

Comment 6: Methods: Please clarify that who the authors reached a sample size of 60 cases.

Authors’ response: Thank you very much. In the Methods section of the manuscript, a new section was added regarding the sample size calculation.

Comment 7: Methods: The statistical analysis section should have sufficient information for reviewers and readers to be able to determine that the methodology is sound and valid for the planned analyses. This section also needs to provide information, such as a power analysis to support the accrual number request, which level of reliability and validity are clinically acceptable. The number of planned subjects to enroll should be adequate to provide sufficient data for validity and reliability results.

Authors’ response: Thank you very much. Detailed info regarding the power analysis was added to the statistical analysis part of the methods section. 

Comment 8: Methods: Please provide the definition of content and convergent validities with supporting references. I strongly suggest the authors to read the following articles before revising the method section of this manuscript. “Assessment of reliability and validity of the adapted Persian version of the Spinal Appearance Questionnaire in adolescents with idiopathic

scoliosis. Spine Deform. 2022;10(2):317-26.” “Guidelines for the process of cross-cultural adaptation of self-report measures. Spine (Phila Pa 1976) 25(24):3186–3191” “Quality criteria were proposed for measurement properties of health status questionnaires. J Clin Epidemiol 60(1):34–42”

Authors’ response: We would like to thank the reviewer for providing informative references to improve the manuscript. Revisions have been made accordingly.

Comment 9: Line 125: what are the self-report scores?

Authors’ response: Thanks. Results of subjective reports are provided in Table 2

Comment 10: Methods: What was the sample size for test-retest reliability?

Authors’ response: Thanks. The sample size did not suffer any changes during the study. Manuscript has been updated accordingly.

Comment 11: Results: Table 1: This table is not provided in a standard form.

Authors’ response: Thanks. Table has been reformatted.

Comment 12: Results: Line 155: Heading of “Consistency and reliability” should be written as “consistency and test-retest reliability”

Authors’ response: Thanks. Heading has been revised accordingly.

Comment 13: Discussion: In the last paragraph of the discussion section, please provide the study limitations.

Authors’ response: Thanks. Paragraph containing study limitations has been added as requested.

---

## [Decision Letter · Decision Letter 1]

24 Oct 2022

Translation, Cross-cultural Adaptation, and Psychometric evaluation of the Persian (Farsi) version of the QoLAF (Quality of Life in patients with Anal Fistula) questionnaire

PONE-D-22-15392R1

Dear Dr. Keramati,

We’re pleased to inform you that your manuscript has been judged scientifically suitable for publication and will be formally accepted for publication once it meets all outstanding technical requirements.

Kind regards,

Zubing Mei, MD,Ph.D

Academic Editor

PLOS ONE

Additional Editor Comments (optional):

Reviewers' comments:

Reviewer's Responses to Questions

**Comments to the Author**

1. If the authors have adequately addressed your comments raised in a previous round of review and you feel that this manuscript is now acceptable for publication, you may indicate that here to bypass the “Comments to the Author” section, enter your conflict of interest statement in the “Confidential to Editor” section, and submit your "Accept" recommendation.

Reviewer #1: All comments have been addressed

Reviewer #2: All comments have been addressed

Reviewer #3: All comments have been addressed

Reviewer #4: All comments have been addressed

2. Is the manuscript technically sound, and do the data support the conclusions?

Reviewer #1: (No Response)

Reviewer #2: Yes

Reviewer #3: Yes

Reviewer #4: Yes

3. Has the statistical analysis been performed appropriately and rigorously? 

Reviewer #1: (No Response)

Reviewer #2: Yes

Reviewer #3: Yes

Reviewer #4: Yes

4. Have the authors made all data underlying the findings in their manuscript fully available?

Reviewer #1: (No Response)

Reviewer #2: Yes

Reviewer #3: Yes

Reviewer #4: Yes

5. Is the manuscript presented in an intelligible fashion and written in standard English?

Reviewer #1: (No Response)

Reviewer #2: Yes

Reviewer #3: Yes

Reviewer #4: Yes

6. Review Comments to the Author

Reviewer #1: MY SHORT COMMENT

I wish to express my happiness for the opportunity to review this manuscript the second time. I indeed went through it and was satisfied with the corrections from the authors. The outcome of the assessment is that the manuscript should be accepted for publication simply because it has met minimum standard for publishing in a reputable journal such as this.

Thank you

Reviewer #2: Dear Authors

Congratulations for your efforts. The manuscript is publishable in this form

Best Regards

Reviewer #3: The authors have satisfactorily addressed all my previous comments and the manuscript is now good to go.

Reviewer #4: The authors addressed all my previous comments and I have no comment on this revised version of the manuscript. Thank you.

7. PLOS authors have the option to publish the peer review history of their article (what does this mean?). If published, this will include your full peer review and any attached files.

Reviewer #1: **Yes: **Chinedu Godwin Uzomba

Reviewer #2: **Yes: **Ozan Volkan Yurdakul

Reviewer #3: **Yes: **Prof Bashir Bello

Reviewer #4: No

---

## [Editor Report · Acceptance letter]

17 Nov 2022

PONE-D-22-15392R1 

Translation, Cross-cultural Adaptation, and Psychometric evaluation of the Persian (Farsi) version of the QoLAF (Quality of Life in patients with Anal Fistula) questionnaire 

Dear Dr. Keramati:

I'm pleased to inform you that your manuscript has been deemed suitable for publication in PLOS ONE. Congratulations! Your manuscript is now with our production department. 

Kind regards, 

on behalf of

Dr. Zubing Mei 

Academic Editor

PLOS ONE